# A Qualitative Investigation of Young Footballers' Perceptions Regarding Developmental Experiences

**Panagiotis Gerabinis * and Marios Goudas** 

School of Physical Education and Sport Science, University of Thessaly, Karies, 42100 Trikala, Greece
* Correspondence: pgerab@otenet.gr

**Abstract:** This study examined perceptions of Greek young football players regarding sport-related developmental experiences using a model of PYD through sport based on results from a qualitative study as a theoretical framework. Twenty one young football athletes (aged 12–15) gave semi-structured interviews. The young athletes identified both positive and negative developmental experiences related to the behaviors of coaches, parents and peers. They did not report any explicit teaching of life-skills. However, young footballers identified their life-skills development by implicit processes. Nevertheless, their understanding of life-skills was rather simplistic.

**Keywords:** positive youth development; youth football; life-skills

## 1. Introduction

The potential of organized sports to promote psychosocial benefits in children has been explored in the last decades. This exploration is widely known as "Positive youth development (PYD) through sport" (Weiss 2016). However, until recently, there has been no deeper understanding of how organized sports programs affect the general development of children and adolescents (Holt 2008; Holt and Knight 2014).

The positive youth development (PYD) framework recognizes children and adolescents as developing structures (Damon 2004; Larson 2000; Lerner et al. 2005) in contrast to older perspectives that youth behaviors are considered as "issues" to be solved. PYD is a dynamic concept based on each person's talents, skills and potential for positive, successful and healthy growth (Armour et al. 2013). Weiss and Wiese-Bjornstal (2009) defined the PYD framework as the development of individual elements and skills, including the cognitive, social, emotional and intellectual qualities necessary for the young people to become functional members of the society. PYD is possible, when these youth then experience the opportunities for skill development through their interactions with significant others such as family, peers, school and the communities in which they live and work. Within this framework, the potential of organized extracurricular programs (Larson 2000; Larson et al. 2006), including sports programs—which programs are the most popular of the organized extracurricular activities (Hansen and Larson 2007; Larson and Verma 1999)—to enhance youth developmental experiences, is widely acknowledged.

To investigate issues related to PYD through sport, two research paths have been followed: The qualitative (i.e., Holt et al. 2017) and the quantitative (i.e., MacDonald and McIsaac 2016). According to Horn (2011), qualitative studies facilitate the promotion of the explanatory approach of PYD.

On the other hand, quantitative research can examine the relations between aspects of organized sports programs and outcomes of PYD, but not the way or the process by which these outcomes occur (MacDonald and McIsaac 2016).

A recent model of PYD development through sport (Holt et al. 2017) grounded on the extant qualitative literature, posits two different processes through which positive developmental outcomes

are gained: (a) An implicit process with the existence of a positive development climate created within the closer sport environment, and (b) an explicit process (involving life skill—building activities and transfer activities) in the presence of a positive development climate. More specifically, a positive youth development climate is determined by the young athletes' quality relationships with the adults (leaders/coaches), peers and parents. Coaches, parents and peers influence youth experience, the first through their central role of influence (Smith et al. 2007), the second through the way they interpret their children's participation in sport (Fredricks and Eccles 2004) and the third, acting as protagonists in a common and specific acceptance context (Smith and D'Arripe-Longueville 2014). The explicit process for the achievement of positive youth development outcomes involves life-skills building activities and transfer activities. Life-skills building activities refer to the use of specific pedagogical and explicit procedures on teaching life-skills. Transfer activities refer to activities that facilitate the application of life skills in non-sport situations. Three categories of positive youth development outcomes are identified: Personal, social and physical ones. The model considers sports programs as microsystems operating within distal ecological systems (e.g. community, culture and policy). Interactions and processes within the sports program microsystem can be influenced by the broader context in which the sports program takes place.

Research on the implicit processes that may facilitate positive youth development in sport, has focused on the perceptions of young athletes and the views of coaches. For example, Harrist and Witt (2015), using focus groups, reported that young female basketball players felt that sport participation facilitated their development in several areas such as social competence, self-confidence and self-discipline and respect. Holt et al. (2009) interviewed young adults regarding their involvement in competitive sport during adolescence. These athletes reported that they learned valuable life-skills, social life-skills, sportsmanship and work ethic through social interactions with fellow athletes, coaches and parents.

Football (soccer) is perhaps the most popular sport worldwide (FIFA Home Page 2006) including Greece. It is therefore somewhat surprising that few studies have recorded young footballers' experiences. Two recent studies have examined the perceptions of young footballers regarding their sports participation. Tjomsland et al. (2016) interviewed young Norwegian footballers aged 12–14 years with a 6–7 years football experience whose teams participated in a large scale intervention project targeting the creation of a positive motivational climate. These authors reported that enjoyment was central among the experiences of young athletes. Young footballers reported that the sense of enjoyment was facilitated by "being with friends, collaborating with teammates, choosing to play the sport, having a supportive coach, and learning new skills, and demonstrating mastery of them." (Tjomsland et al. 2016, p. 836)

A qualitative study by Santos et al. (2018) examined the perceptions of young footballers and coaches, regarding how competitive youth football may facilitate PYD. Nineteen young Portuguese footballers aged 13–15 years with 3–10 years football experience who participated in competitive leagues were interviewed alongside four coaches. Their results showed that although the coaches had good communication with their athletes, they did not seem to use clear practices or any explicit approach to facilitate PYD, and that the performance outcomes, sometimes, superseded the PYD outcomes. These authors concluded that coaches need specific knowledge to promote positive development and how to use their positive communicative in a competitive youth sport.

The present study extends the research of Santos et al. (2018) and examines the experiences of young Greek footballers using the model of Holt et al. (2017) as a theoretical base. Specifically, the study examines the perceptions of young footballers regarding to the following components: The role of peers, parents, adults in structuring a PYD climate, life-skills program's focus and PYD outcomes.

Given the relatively small number of studies that have examined young athletes' perceptions regarding sport-related developmental experiences, and particularly in soccer, it is important to understand further the impact of sports participation from their own point of view. As a non-intentional, implicit path leading to PYD outcomes has been suggested (alongside an explicit one, Holt et al. 2017), these young athletes' perceptions are crucial in gaining further insight in this process.

## 2. Methodology

### 2.1. Research Questions

Three research questions based on the Holt et al. (2017) model drove this study: (a) How do young athletes perceive the role of adults (coaches), parents and peers in youth football? (b) Do young footballers feel that they can develop life-skills through youth football, and how? (c) What positive youth development (PYD) outcomes do young athletes think they can attain through youth football?

### 2.2. Participants

Twenty one young football athletes (aged 12–15) from nine different youth sports clubs in a city of Central Greece participated in the study. A purposeful selection procedure was followed. The coaches of the sports clubs were asked to denote two athletes, one highly motivated and one not so motivated. Further, these athletes had to be engaged in football for at least 5 to 6 years. The study was approved by the Bioethical Committee of the University.

### 2.3. Context of the Study

Sports clubs have the responsibility for the operation of local youth football within the regulations of the Local Football Association which operates under the umbrella of the Greek Football Federation. Training and games are carried out by the graduates of University sports departments, usually with a specialty in football, or by coaches of UEFA schools operating under the responsibility of the Greek Football Federation.

In this context, the local association organizes championships for under-16, under-14, under-12 and under-10 age categories. The scoring process of these championships is posted on the official site of the local association. For each age category, there are specific regulations. The best players in the U-16 and U-14 championships represent the teams of the local association participating in the national championships.

### 2.4. Methodological Procedure and Information Collection

Twenty one semi-structured interviews were conducted, lasting on average thirty five minutes. In particular, 4 interviews lasted 20 to 30 min, 2 interviews 30′–35′, 8 interviews 36′–39′ and 7 interviews 40′–46′. The interviews were taken during November and December (winter), which is around mid-season in Greece. Interviews took place in the offices of the sports clubs, in changing rooms, and in some cases in the athletes' homes. A week before the interview, each athlete received a letter informing about the purpose of the study and ensuring anonymity. Approvals were given by all of the parents of the children. In order to understand the interview process better, each child received a short questionnaire for his experiences in football, in relation to his proximal environment (coach, parents, peers). Each child returned the questionnaire on the day of the interview.

### 2.5. Data Analysis

As the research was based on a particular theoretical model (Holt et al. 2017), a deductive thematic analysis was used. Thus, the components of the model served as predetermined higher-order themes.

The analysis of data took place in six phases (Braun and Clarke 2006): In the first phase, the researcher made the transcription and read the text multiple times. Secondly, he identified and coded the text segments and categorized them in the predetermined higher-order themes.

Thirdly, the higher-order themes were broken down into second-order and first-order themes. Fourthly, the themes were reviewed in relation to the coded extracts, and new text segments that could not be grouped into the predetermined higher-order themes were identified. Fifthly, the researcher defined and named the final themes and created the table with all the coding data. At the final phase,

the report was produced. All of the above "steps" were constantly checked by a second academic researcher who was experienced in qualitative research methods.

*2.6. Establishing Trustworthiness*

According to Lincoln and Cuba (1985) four criteria are necessary in order to ensure the trustworthiness of a survey: Credibility, transferability, dependability and confirmability. In terms of credibility, the first researcher has been a youth football coach for fifteen consecutive years, and is familiar with the procedures in this field. Regarding the transferability of the research, the clubs where the children are practicing are typical football clubs operating under the direction of the local football association, and by extension, the Hellenic Football Federation. Regarding dependability, all the children who participated in the research had been engaged with football for at least five years. Confirmability was supported by the continuous supervision of an experienced academic researcher in qualitative analysis. The confirmability of the interview process was also supported by the information letter and the short questionnaire that were given to these young players before each interview.

## 3. Results

The transcription of the interviews yielded 258 single space pages. Data analysis established the predetermined higher-order themes of "Parents", "Peers", "Coaches" and "PYD Outcomes". "Life-Skills program focus" was replaced by the theme of "Life-Skills development". Young athletes' views on the role of sports clubs in youth football did not fit in the predetermined higher-order themes, and formed a new higher-order named "Sport Club". Also, quotes related to the negative experiences of young athletes, formed another higher-order theme named "Negative Outcomes-Experiences". Table 1 summarizes the themes that emerged in the data analysis. Below, these themes are described alongside their respective quotes.

*3.1. Coaches*

The first higher-order theme of "Coaches" includes the themes of "Winning focus" "Learning focus", "Positive climate" and "Negative behavior". The theme of "Winning focus" was categorized further into outperforming-result, unequal participation and pressure. Young footballers identified a number of winning focus coaches who were more interested in the result of the game: "He is more interested in winning; he doesn't care so much what we will learn in training, but only in the outcome of the games". These coaches usually use the best players during the game, while the different skill levels of the players causes unequal behavior towards the athletes:

> Those children who don't do so well, he uses them, but not so much, but he will put someone stronger, ... to retain the positive result of the game, and those children who are weaker, he will allow them to play, but he will use them only for fifteen minutes. (Interview 7)

Also, intense pressure is evident in some coaches' actions: "Once, instead of pushing the ball to the side soccer line, I pushed the ball to the corner line ... he pulled me out ... he began to shout and he told me not to come for the next day's training". Additionally, the negative comments and negative behavior of some coaches were the result of young athletes' mistakes: "He was throwing things on the kids, he had a stopwatch and he was throwing it on the kids ... or he was jumping up and down furiously".

A number of coaches with "Negative behavior" were identified. This theme consists of three first-order themes: Shouting, negative comments and conflicts. Young athletes speak about coaches who shout at the referee when they feel the team is been wronged: "He was shouting, many times. That is, sometimes when there was a foul and it seemed to be a foul, he was shouting to the referee and sometimes he was reacting negatively". The young athletes did not provide examples of their coach's negative behaviors only, but also of those coaches of other teams:

In a recent game, at half-time, both teams entered into the locker rooms, and we are listening to the other coach who was shouting at the kids too strictly and too loudly. And I believe that scares the children and discourages them. (Interview 6)

In some teams there are very irritable [coaches], they take everything seriously, and they don't understand that we are children and we play to be happy, and sometimes they accuse the children or the referee if he makes any wrong decision or anything that is against his team. Generally some coaches are not the appropriate role models for their athletes. (Interview 4)

**Table 1.** Data coding.

| Higher-Order Themes | Second-Order Themes | First-Order Themes |
|---|---|---|
| Coach | Winning focus | Outperforming-Result |
| | | Unequal behavior |
| | Negative behavior | Pressure-Negative behavior/comments |
| | | Shouting |
| | | Negative comments |
| | | Conflicts (e.g., With other coaches/referees) |
| | Learning focus | Participation |
| | | Focus on improvement |
| | Positive climate | Coach listens-Concern |
| | | Warm climate, Calm-Patient |
| Parents | Concern-Communication | Interested for their effectiveness |
| | | Participation encouragement |
| | Lack of interest | Due to school duties |
| | | Due to non-participation in games |
| | Negative Profile | Conflicts |
| | | Negative behaviors to other kids |
| Peers | Relationship development | Encouragement |
| | Negative behavior | Negative Attitudes. Negative behaviors in games |
| Sport Club | Winning focus | Focus to win-Unequal participation |
| | | Unsporting behavior |
| | Learning focus | Athletes' Participation-Team spirit |
| | Positive atmosphere | Quality relations |
| Life-Skills development | Initiative development | Coach gives chance for action |
| | Goal Setting | Effective play |
| | | Winning |
| | Cooperation – Teamwork | Advice |
| | | The sport |
| | Concentration | Shouting-Punishments |
| | | Verbal advice |
| | Problem solving | Handling defeat |
| Positive youth development (PYD) outcomes | Outcomes related to coach's actions | Competence-ImprovementPositive feelings |
| | Outcomes related to football | Personal attributesDecision making (to manage situations) |
| | | Emotional management |
| | | Friendly relationships–Sociability |
| Negative outcomes/experiences | Coach | DropoutNegative Emotions |
| | | Negative model–Unsporting behaviors |
| | Sport club | Negative behaviors–Negative emotions |
| | Football in general | Intense–Negative emotions |
| | | Injuries |
| | Parents | Negative image |
| | | Pressure–Anxiety |
| | | Unsporting behavior |

On the other hand, there are coaches' actions that fit within a higher-order theme of "Learning focus". This comprises two first-order themes: Participation and focus on improvement. Firstly, there

are coaches who offer participation to all the children: "In the games, he ... doesn't look at who is better and who is worst, but he offers equal participation to all of the children", "I believe, that he offers [participation] to all the children not only to the best, but when it is possible, he uses everybody, as he did in the last game". In addition, there are coaches who are interested in the improvement of their footballers, and they try to encourage their athletes' efforts:

> Basically when we make some mistakes he stops us in the middle of training and explains to everyone differently. And not just to the player who did it, but to the whole team because we could have made the same mistake too. (Interview 1)

Finally, there are coaches that create a "Positive climate" and they listen (they are interested in the children's opinion), they show concern and they create a climate of warmth:

> Once, when I didn't feel well with my position during a game, I told him to change me, so he changed me and then I felt much better, and more comfortable, and in general, in training, if a child wants to say something, he listens to him. (Interview 4)

> He ... tries to ... find a good position for everyone, everyone to play and to enjoy, and ... generally tries to make some changes [in positions] for some kids, to feel better, to be more pleased with themselves, he makes jokes, he has a lot of humor. He will try not to stress us too much, but he tries to enhance our positive psychology. (Interview 14)

These coaches also show calm and patient behavior and they don't shout:

> My coach is a good coach, he works with us, he doesn't shout, he gives us advice to become better, and ... when he doesn't like something, he doesn't speak to us nervously, but he tries to speak nicely and calmly for us to understand it. (Interview 12)

The results denote that in most cases those coaches who have a learning focus do create a positive warm climate: "The coach gives more emphasis on the playing field, he wants us to have good co-operation, to play as good as we can and he gives more emphasis on participation. He is not interested too much in the result of the game". In contrast, young athletes report that coaches who focus on winning rather than learning exhibit negative behaviors: " ... He had told us once, that if we didn't win the game, we would run in all the court". " ... He shouted too much ... he didn't believe in us, he wanted only to win and nothing else".

### 3.2. Parents

The higher-order theme of "Parents" includes the second-order themes of concern-communication, lack of interest and negative profile. In terms of "Concern-Communication", some children report that their parents are interested in their personal and team effectiveness:

> When the game ends and we get into the car, he tells me in the end ... I ask him how I played, he tells me honestly, then we talk in general about how we all played, the team and ... we talk about it in general, and at home for a while. (Interview 4)

Young athletes report that some parents are bonding with them, support them and encourage their participation in the sport: "Yes. He encourages me many times, all these years; he likes to be involved with me and generally encourages me in difficult situations".

> "Okay, he will not tell me if I lose or I win. He will congratulate me if I win or I score, but if I lose, he will not tell me why l lost, he will tell me OK, it does not matter, good ... you tried, and you played well". (Interview 19)

Regarding the "Lack of interest", some children report that their parents are not generally interested in football, considering that the young athletes are missing time from their school duties.

Some young athletes also report that their parents discourage them from continuing to play football, because they do not play in the games.

Concerning the "Negative profile", children comment on parents of other children who are involved in quarrels and speak badly to others, such as coaches, other parents and referees:

> Now in a recent game, a referee ... as far as a decision is concerned: It was a foul? A penalty? I don't remember ... Some parents, two or three, were swearing at him in a vulgar way ... and that's not good because the referees are new guys and they do their job. (Interview 6)

There are also several reports of negative parental behaviors towards young footballers (e.g., admonishments, shouting and pressure): "I've seen a parent to be obsessed with his son, shouting out of the field and affecting his son very much and telling him ... "do that, do that, hit him" and his son ... to hit [the others players]". Below another distinctive example, follows:

> A child, I do not remember who he was, he did not play well in a game, and his father was very [obsessed] ... about football and how this kid should play, he wanted him to play perfectly, not to make mistakes, not to make stupid things and ... once he did not play well in a game, left his son there to go home alone. Or another time he put him into the trunk (of the vehichle) and they returned home. (Interview 19)

### 3.3. Peers

The third higher-order theme of "Peers" includes the second-order themes of relationship development and negative behaviors. As far as the second-order theme of "Relationship Development" is concerned, there are a number of young athletes who report that in their team, they encourage and inspire each other:

> Yes, many times, in the games especially, if an error occurs, we will say it doesn't matter, we will not blame anyone, or if one does something good, we will tell him well done, to encourage him, and generally we try to support each other. (Interview 14)

Also, the concepts of teamwork and good relationships are evident in some of the children's references: "In the team all the kids are friends ... We generally have a warm atmosphere in the team and rely on each other because we have developed a relationship for two or three years".

> More in the games, but also in training ... The only thing I see, showing the team spirit, is that ... for example when someone will make a mistake, we will all run to help and we will not tell him what you did and why, we will encourage him not to do it next time and we will make an effort all together. We will not discourage the one who made the mistake (Interview 4)

"Negative Behaviors" are denoted by children who report the negative attitudes of other young players, e.g., for comments and negative behaviors in their team: "Generally, when there is a mistake in the team, for example in the defense, and the team concedes goals, they will ... shout against the defender. I've seen this many times". Negative behavior is also related with aggressive and unsportsmanlike behavior in games. For example: "They were beating the opponents on purpose and cursing at them, they cursed the referee and the fans of the opposing team [his team] and when the opposing team lost, they laughed at them".

### 3.4. Sport Club

The fourth higher-order theme "Sport Club" includes: Winning focus, learning focus and sports clubs with a positive atmosphere. That is, some children report that there are clubs that are primarily interested in the outcome, and that there is an unequal participation of children in these teams:

> "For example, there is a club that wins the championship every time, but some kids don't play, they don't take participation time, just to hear a rumor that, oh! This academy [club]

gets the championship and is the best". By extension, young footballers state clearly that there are clubs that favor unsportsmanlike behavior: "Yes, some teams have the philosophy to play more aggressively, and I've seen some others [players] fight, but it does not happen to us".

In relation to the "Learning focus" clubs, some athletes report the promotion of the participation and teamwork: "The coach and not only him but generally this club, does not try only to . . . win, but also, they want the children's participation". A "Positive Atmosphere" is apparent in clubs that develop quality relationships among peers:

> The atmosphere we have here [at the sports club], has helped us think that even outside the academy, there will be one, two, three of my peers that we'll be able to talk about something else, we'll be able to have a friendship and they'll always be there for me, and this kind of relationship we have there isn't fake. (Interview 4)

### 3.5. Life-Skills Development

The fifth higher-order theme relates to the development of life-skills. To frame this part of the interview, the participants were provided with a description of the term "life-skills" based on the respective definition by Danish et al. (2004). Further, in order to focus on the processes of life-skills development, the participants were asked to comment specifically on the development of initiative, goal-setting, cooperation-teamwork, concentration and problem solving.

Young footballers did not mention the explicit teaching of life-skills and life-skills transfer, in a sense of intentional and intergraded teaching within the sports program. However, they reported a "reactive" teaching of life skills as well as the "by product" development of life-skills. Bean and Forneris (2017) have described "reactive" life skills teaching, as it is not intentionally planned, but occurs after a respective opportunity, such as for example, a conflict between two athletes and 'by product' development of life-skills when a coach does not use respective intentional strategies, but a secondary result is a life-skill produced.

Regarding "Initiative," some children report that their coaches give them the opportunity to take some initiatives on the football field: "Yes, for example, last year, when we were a quartet in the center, he told us that when you want to change a position, you don't have to ask me, I trust you". Also, several players mention that coaches give the possibility for initiative under certain conditions, but they don't give a full freedom of movement:

> In the games he tells us most of the time what to do, but if a player does the right thing on his own initiative, he doesn't say anything to him. But if he does something wrong, the coach shouts. (Interview 3)

Goal setting consists of two first-order themes: Effective play and winning. Footballers who recognize goal setting as an improvement in football effectiveness refer to learning oriented coaches:

> Our coach does not aim to win the championship, nor win, he wants us to play as a team, and . . . I can say he wants us to have the possession of the ball and he tells us that . . . when we have the possession, to play in our square in a simple way, and in the other square to take risks. (Interview 5)

Secondly, young athletes recognize as goal setting the pursuit of the victory, the winning result, and the winning of the championship: "The championship. We want to win for two or three years. Since I came to this team, we have not won a championship, and we are in the second place, and we are continually aiming for it".

The third life-skill of "Cooperation-Teamwork" is considered to be a result both of the coach's advice and through sport itself. As far as the first-order theme of advice is concerned, some coaches ask their players to cooperate and to have more team spirit: "In every training session he advises us

. . . a child asked him why it is important to pass the ball . . . and he explained that teamwork is the best way to play a game".

As far as the first-order theme of sport, children believe that soccer as a team sport can promote cooperation: "Football is a sport, that if you don't collaborate . . . in the game, you will not win or do well".

The next life-skill of "Concentration", is considered to be enforced by shouting or by verbal advice: "Yes, in the training when the children talk to each other, the coach shouts at them, and generally when they are not focused during training . . . he shouts". Moreover some children say that their coaches request concentration, verbally: "He speaks to us before the training and before the game, that . . . this hour that you came here, you have to be concentrated here, and if anything else occupies your mind, you can think of it later".

Finally, when the young athletes were questioned about "Problem solving" they referred to handling defeat. They spoke about their team spirit created in hard times, for example when losing a game, and that they are facing these situations through team unity: "We don't blame anyone particularly [in losses], but we are together, we operate as a team, . . . when one makes a mistake, even if this mistake brings the loss, no one tells him anything".

> Children, who don't have anxiety, encourage those who are stressed, that is . . . you will play well, do not get stressed, and if we lose it doesn't matter, we will not die . . . okay. It's just a game, we are telling each other. (Interview 19)

### 3.6. PYD Outcomes

Two second-order themes comprise the higher-order theme of "Positive developmental outcomes". Outcomes related to a coach's actions and outcomes that are related to football. Firstly, some young athletes note that the improvement in individual competence is due to the coach's contribution: "He is very good, he doesn't shout, and through his training we learn a lot of things. Many children have improved with his help"

> Sometimes we do special exercises on how to improve each one individually. When . . . a child does the exercise well and manages it, his performance will rise, because his confidence has also risen, so he will be able to perform better. (Interview 4)

In addition, some athletes point to the positive feelings that result from the coach's actions:

> In the previous team . . . when I was playing at center back and I was making some good actions, for example, a good defense or a good pass in the area, the coach was saying, well done, and my psychological mood was getting better. (Interview 11)

The opportunity of participation in competitive games also creates pleasant feelings: "I enjoyed this once . . . at this year. It was the first time. I started as a basic player in my team. And I enjoyed it very much. And we had a good result".

Regarding the second-order theme of "Outcomes related to football", the young athletes report that through youth football they developed attributes such as self-confidence, self-awareness and persistence: "[I've learned] . . . To try for the best, to set goals, to try day by day and not to fall, for example, when I have a failure, to continue to train hard and to try to reach my goals". Furthermore, children report that they have gained benefits (e.g., discipline, persistence, organization, self-confidence, etc.) through football, that help them to manage situations: "From football you learn to push yourself, to improve yourself, to organize your time better, to know when to do this, when to read this . . . But much more to push yourself to become better". Young athletes also reported some experiences/outcomes related to emotional management. "With other people . . . I'm a bit more patient, I wasn't before, and . . . through football and the games . . . I learned to be patient and not give up directly and not to get upset like I did". Additionally, for some children, football is a sport that fosters friendly relations and sociability:

Teamwork, patience with others, being good, not being . . . tough, to care about, because if you are bad with a teammate, and . . . you affect his psychology, so, the same goes with out of soccer, if you are bad and strict with a friend, you will make him feel bad. (Interview 6)

*3.7. Negative Outcomes—Experiences*

The final higher-order theme of "Negative developmental outcomes—experiences" is categorized in the second order—themes of Coaches, Sport Club, Football, Parents and Peers. First, some coaches' behavior results in negative emotions and dropouts, and athletes' unsporting behavior. More specifically, some young athletes report that they and their peers have experienced negative emotions such as fear, discouragement, dissatisfaction and frustration due to coaching behaviors: "When we had this coach, when the kids were having the ball, they were afraid and they would give the ball straight back, in order not to make a mistake, and the coach starts shouting". "[Discouragement] . . . Sometimes, in games, but also in training, when something is going wrong, he shouts, and then we do it worse". "[Frustration] . . . Sometimes if I make the same mistake . . . a wrong pass, he would tell me that I did badly, even though the court was in a poor condition. And that bothered me a lot". Moreover, there are some reports about children who dropped out or for children who were thinking to stop playing football because of the coach's behavior:

In the team I remember only one child that he left because the coach was shouting at him very much, because he didn't have a good technique and he was making a lot of mistakes, no . . . two children left. And they left, they left football completely (Interview 21)

Finally, the negative behavior of the coaches creates a negative image in the eyes of children, and these coaches are negative models who create unsporting behavior among peers: "Yes, the coach was not the best, he was swearing during the game, and the referee had pointed it out to him, but . . . the children did the same as the coach did". "They were afraid of the coach and they were playing harder in case he shouted at them. The players were more committed to the game, and they were playing hard, they were beating . . . "

There are also reports of the "problematic" attitudes of specific sport clubs that fostered inappropriate behavior as well as negative feelings: "Some children were also influenced by previous teams [they had played before] and they were playing very hard, they were cursing, but they got these behaviors from those teams". "At that academy, then I realized fear. In previous academies I was not feeling either fear or anxiety".

Moreover, some children believe that "Football in general" causes negative emotions and injuries. Regarding negative emotions, they report aggression and dislike: "They are out of line [peers] . . . I believe they are affected by . . . the intensity of the game, and they become inappropriate". "Football can make you hate someone because he is more selfish, he doesn't give passes or lose a phase" Secondly, some athletes reported injuries as a negative outcome from youth football:

I worry when my teammates or guys from other teams are injured. This makes me sadder, because if someone is in his position, he feels that he falls back, he can't play with his teammates, and that disturbs me more. (Interview 7)

Parents' behavior results, sometimes, in pressure-anxiety, and in unsporting behavior. A young athlete reported that: "I had the experience to see a parent of the opposing team who has entered the field at the half time of the game, and calls names at his son so hard that he began to cry". Some interviewees also reported the feeling of pressure from parents. For example: "Will he speak badly? He will tell him words that will stress him, will affect him psychologically, just . . . I feel that sometimes some parents talk without . . . knowing football". Also, some reports describe parental negative behavior that creates tension in young footballers and involves them in hostile and unsporting behavior: "Yes. Yes. I've seen it. It was . . . a father who was talking badly to the referee, and his child was very aggressive, and he was constantly hitting and swearing".

Peers' behavior is another factor of negative experiences. More specifically, there are cases where the relationships between peers cause tensions: "They separate the skillful players from the not so skillful. And it is like an intimidation because ... with every mistake they find the chance to say something". "A lot of guys show their bad self, they curse, beat, play very hard and often they humiliate our opponents and their own teammates."

> They bother other children who are not so good. A child who is not a good player, he doesn't know dribbling, and he tries to get into the team, but some kids drive him off, and they tell him that he is not playing well, and to go to another team. (Interview 16)

## 4. Discussion

This study aimed to examine the perceptions of Greek young football players regarding their developmental experiences, following the model of Holt et al. (2017). Specifically, young footballers were interviewed about the attitudes and behavior of significant others in youth football, such as coaches, parents, peers and sport clubs. Further, their perceptions regarding life-skills and competencies they developed through football participation were elicited.

Young athletes identified two types of coaches. Coaches with a winning focus give a lot of pressure to players, react to players' mistakes with negative comments and often have conflicts with referees and other coaches. On the other hand, coaches with a learning focus are more interested in their athletes' progress and participation, and create a positive motivational climate These results show that young athletes were able to identify coaches whose actions could be described in terms of the theoretical concept of two types of coach-related motivational climate—an ego or performance involving and a task or mastery involving one (Duda and Nicholls 1992; Smith et al. 2008). As the vast majority of studies on the sports motivational climate have followed a quantitative approach with the employment of questionnaires, the present study enriches this literature by providing more details regarding respective coaches' behaviors through a qualitative approach. Given the multitude of studies that have emphasized the positive motivational effects of a mastery motivational climate and the detrimental motivational effects of a performance-oriented motivational climate, (see Harwood et al. 2015 and Papaioannou et al. 2012 for reviews), this study shows that young footballers may have positive or negative experiences related to the motivational climate created by the coach with respective effects on their motivation.

Young athletes also pointed out that the overall philosophy of the club affected the coaches' approach, as well as the behaviors of parents and peers during practices and competitions. As young athletes commented not only for the philosophy of their club, but also for the overall approach of other clubs, it can be stated that sport clubs of the same league constitute a decisive element of the distal ecological systems component of the Holt et al. (2017) model of positive youth development through sport. Holt et al. (2017) stated that distal ecological systems comprises, among others, community, policy and culture. This study adds to this component of the model an additional factor, the overall philosophy of the sport clubs. Additionally, these results attest to the limited respective results of previous studies that have highlighted the effects of the organization's overall policy and approach on young athletes' development (Holt et al. 2008; Fraser-Thomas and Côté 2009).

The interviewees reported parental behaviors that show interest and communication for both individual and team effectiveness, as well as their support for children's active participation in football. However, there are parents who exhibit negative behaviors mainly during the games, as they shout or speak badly to children, coaches and referees. Also, some parents show little interest or discourage their sport engagement either because of school duties, or because they do not participate in games. The results confirm the positive and negative behaviors of the parents mentioned in the literature, for example for parents who encourage and support their children's athletic activity (Fraser-Thomas and Côté 2009), but also for parents with negative and stressful behavior which is influenced by the pursuit of the result (Gould et al. 2008).

Strandbu et al. (2017) reported that young athletes desire their parents' presence but, they don't like them to be involved in their relationships with their peer athletes. Mixed views of young players regarding their peers emerged. On the positive side, there were comments related to the development of team climate and more generally to relationship development, encouraging and inspiring each other. Similar accounts have been offered by Bruner et al. (2017) regarding young ice hockey athletes who felt that pro social interactions with their teammates enhanced their social identity. Also, Fraser-Thomas and Côté (2009) noticed that through swimming the young athletes identified opportunities to develop close and unique friendships built on common interests. On the other hand, there are reports of negative comments among the team members, mainly due to the mistakes made in the games. Keegan et al. (2009) identified "pressuring behaviors" among peers in case of wrong decisions.

Moreover, in the present study, some of the young athletes reported unsportsmanlike and aggressive behaviors of other team athletes during the games. According to Smith et al. (2006) young athletes who are influenced and motivated by the performance, develop self-centered thinking and they see others as opponents, and they develop conflicts.

Regarding life-skills development, the young footballers in this study reported that spontaneous and reactive teaching by the coach as well as the game of football itself leads to the development of life skills. These processes of development fit in the "reactive" and "by product" processes described by Bean and Forneris (2017). Further, the young athletes did not mention any explicit teaching of life skills by their coaches and any teaching for transferring life skills in domains other than football. The young athletes felt that football itself helped them acquire skills, however they did not mention any use of these skills in other contexts. The results also showed that young athletes have a "naïve" view of life-skills which is rather different from the concept of life-skills as this is outlined in the literature. For example, theoretically, acquiring the skill of goal-setting should involve an understanding of the principles of effective goal setting, and applying these in relation to training and competition, whereas in this study, young footballers seem to equate goal setting with a long-term goal such as being the champions in the league. Clearly, in this age group, the explicit teaching of life skills, in the examples of programs such as GOAL and SUPER (Danish and Nellen 1997) is warranted in order to maximize the social-cognitive gains of sports participation.

Regarding PYD outcomes, young athletes reported a range of personal attributes that were developed through their participation in football. This implies an implicit path of positive youth development as depicted in the Holt et al. (2017) model. Similarly, previous studies (Kendellen and Camiré 2015; Fraser-Thomas and Côté 2009; Jones and Lavallee 2009) showed that athletes felt that the sport itself taught them valuable life skills applicable in other contexts.

## 5. Conclusions

This study showed that Greek young footballers have both positive and negative experiences through their interactions with coaches, parents and peers, not only within the microsystem of their own club, but also within the microsystem of other football clubs. The philosophy and policy of football clubs emerged as a distinct element of the distal environment. The young athletes who participated in the study did not have any experience of explicit life-skills teaching. However, they felt that participation in football had helped them develop a range of positive personal attributes. A limitation of this study is that the deductive analysis employed may have restricted the depth of the description of young footballers' lived experience.

**Author Contributions:** Panagiotis Gerabinis and Marios Goudas contributed equally to all parts of the manuscript.

**Funding:** This research received no external funding.

**Conflicts of Interest:** The authors declare no conflict of interest.

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
