# Peer review of "A Qualitative Investigation of Young Footballers’ Perceptions Regarding Developmental Experiences"

_socsci, doi:10.3390/socsci8070215_

Reviewer 1 Report

This is an interesting paper and has all the ingredients for a research paper. Having said this the paper requires some restructuring before it is published. There are some key issues that need addressing:

At the moment the results come before the methods, for flow and clarity, the methods need to come after the introduction and before the results. The methods need some clarity, was there, for example any need to consider how different clubs may impact on PYD?

Holt's PYD model needs greater discussion in the introduction - at the moment it is discussed too fleetingly and it needs to be expanded upon.

The focus of the paper on Greek players needs to be in abstract and much earlier on in the introduction

The contribution to knowledge needs to be identified, the authors note it extends the research of Santos (2018) but why is this significant?

In the discussion the paper does not locate the findings of their research in the extant literature and this needs addressing, how does the results of this study relate to other research?

The conclusion needs to be redrafting, what are the key findings, what do these findings contribute to knowledge? These need summarising as well as an acknowledgment of the limitations of your findings.

referencing style needs checking.

Author Response

We would like to thank both reviewers for their overall positive evaluation of our paper and for providing us with the chance to work further on improving and strengthening our manuscript based on valuable comments.

We have now revised our manuscript addressing and responding to all the comments provided. Please find below our detailed responses to each specific comment. We have also carefully checked the manuscript for typographical or grammatical errors making appropriate revisions.

As suggested by the editor we have used the Track Changes option in revising the manuscript, thus all new materials and amendments have been highlighted with  blue red-colored text while in our responses in each reviewer’s comment we have provided the page and the lines of the respective revisions within the revised manuscript. Please note that the page and lines provided for the amendments are before Changes are accepted.

We believe that our revisions have addressed all your and reviewers’ comments and have improved the quality of the manuscript. Thus, we hope for a favorable light on our revised manuscript.

 Open Review   (1st Reviewer)

Comments and Suggestions for Authors

This is an interesting paper and has all the ingredients for a research paper. Having said this the paper requires some restructuring before it is published. There are some key issues that need addressing:

Thank you for commenting positively on our paper and for providing the opportunity to improve the manuscript.

At the moment the results come before the methods, for flow and clarity, the methods need to come after the introduction and before the results.

We have now rearranged the content accordingly

The methods need some clarity, was there, for example any need to consider how different clubs may impact on PYD?

This is an interesting comment. However, respectfully, we have not examined the influence of specific sport clubs on PYD outcomes. There are two reasons for this: a) Our aim was to describe young footballers’ experiences. As such it has been somewhat out of the purpose of the study to examine “cause and effects” regarding PYD. B) The young athletes provided experiences not only for their own club but also regarding their interactions with coaches/parents/athletes of other clubs. Therefore, their positive or negative experiences were not solely the result of their participation in their own club. 

Holt's PYD model needs greater discussion in the introduction - at the moment it is discussed too fleetingly and it needs to be expanded upon.

We have now described the model in more detail. New materials added can be found on p. 1-2  l. 40 - 70

The focus of the paper on Greek players needs to be in abstract and much earlier on in the methodology

We have added this information in the abstract and on p. 1 l. 6-12 and on p. 3, l. 141-157

The contribution to knowledge needs to be identified, the authors note it extends the research of Santos (2018) but why is this significant?

Thank you for pointing out this important omission. We have added a respective statement regarding the significance of the study (p 2, l. 90-107)

In the discussion the paper does not locate the findings of their research in the extant literature and this needs addressing, how does the results of this study relate to other research?

We have followed this comment and we have redrafted the discussion to address the relevance of our results to other research

The conclusion needs to be redrafting, what are the key findings, what do these findings contribute to knowledge? These need summarising as well as an acknowledgment of the limitations of your findings.

We have followed this comment and we have redrafted the conclusion accordingly.

referencing style needs checking.

Reviewer 2 Report

Thank you for the opportunity to review the manuscript entitled, “A qualitative investigation of young footballers’ perceptions regarding developmental experiences.” I found the manuscript to be well written and the study to be informative and insightful for the literature in positive youth development through sport. I have made a number of suggestions below for the authors to consider, with the goal of enhancing the potential contribution to the literature.

Introduction:

-          The authors introduce the PYD approach and present the Holt et al., (2017) framework (L. 40-49). It would be beneficial to explicitly describe all components all of the model (i.e., Distal Ecological Systems), to provide a deeper foundation to present the findings and conclusions of the study.

-          L.58-69. Two studies are introduced to provide detailed context for the study of PYD in youth footballers. This section would be enhanced if more specific information is provided about the context of each study. For example, adding the age-group, level of competition (e.g., club, competitive), location of the sample in each study.

Results:

-          The findings are presented provide interesting insight about the developmental experiences of youth footballers. However, as Holt and colleagues model provided a meta-synthesis of existing knowledge about PYD in youth sport, I believe that the deductive analysis as presented does not provide the potential depth that it could. That is, only presenting the categories from the model provides confirmation of the model. The greatest insight from the findings centers around the ‘PYD outcomes’ and the ‘Negative Outcomes/Experiences’ as the themes appear to focus on the interaction between the individual athletes and the context. Could these be emphasized further to show how the findings inform specific components of the model?

o   The themes, as currently presented, highlight both positive and negative experiences and outcomes. Delineating these more explicitly would help to show what/how PYD occurs in the youth football context

-          L. 197- How was the ‘Sport Club’ theme identified in the deductive framework? Does this represent the ‘Distal Ecological System’?

-          L. 215 - The ‘Life Skills’ theme and sub-themes presented specific developmental outcomes, that in my view, would align more clearly under the ‘PYD outcomes’ theme? Holt et al., present the implicit process and the explicit process, with an emphasis on the learning process. Explain this process in addition to the specific PYD outcomes would be helpful to realign this section to the model may help to show this connection.

-          The authors state the ‘young athletes have a “naïve” view of life skills” (L.218-219). This is an interesting statement that could be expanded on – does this mean that life skills are not being learned to the degree explained in the literature?

Discussion:

-          The authors describe “two types of coaches” (L. 343-). This highlights an interesting finding that adds to the literature of coaching for PYD. However, it would have been beneficial to present this more clearly in the results section to clearly delineate, with quotes and themes, how the two types of coaches behaviorally differed.

-          The authors state that the explicit process was not mentioned by participants (L. 368-). Does this infer that the process is not necessary and that implicit development of life skills occurs? What recommendations can be taken from such findings? Adding the specific practical recommendations for coaches would add depth to the paper.  

Methods:

-          The choice of a phenomenological approach is based on an ontology of multiple perceptions of reality and an epistemology of subjectivity of human knowledge, yet is presented in a way that emphasizes “common meaning” (L. 411) and aligned with a deductive, theory-driven analysis. This appears to be an odd pairing of methodological approaches and a lack of methodological coherence. Can the authors explain and clarify the fit of these approaches?

-          Adding detail about the timing/length of the interviews would be helpful.

Author Response

 We would like to thank both reviewers for their overall positive evaluation of our paper and for providing us with the chance to work further on improving and strengthening our manuscript based on valuable comments.

We have now revised our manuscript addressing and responding to all the comments provided. Please find below our detailed responses to each specific comment. We have also carefully checked the manuscript for typographical or grammatical errors making appropriate revisions.

As suggested by the editor we have used the Track Changes option in revising the manuscript, thus all new materials and amendments have been highlighted with  blue red-colored text while in our responses in each reviewer’s comment we have provided the page and the lines of the respective revisions within the revised manuscript. Please note that the page and lines provided for the amendments are before Changes are accepted.

We believe that our revisions have addressed all your and reviewers’ comments and have improved the quality of the manuscript. Thus, we hope for a favorable light on our revised manuscript.

Open Review (2nd reviewer)

Comments and Suggestions for Authors

Thank you for the opportunity to review the manuscript entitled, “A qualitative investigation of young footballers’ perceptions regarding developmental experiences.” I found the manuscript to be well written and the study to be informative and insightful for the literature in positive youth development through sport. I have made a number of suggestions below for the authors to consider, with the goal of enhancing the potential contribution to the literature.

Thank you for your positive overall evaluation and for providing us with the opportunity to work further with the manuscript.

Introduction:

The authors introduce the PYD approach and present the Holt et al., (2017) framework (L. 40-49). It would be beneficial to explicitly describe all components all of the model (i.e., Distal Ecological Systems), to provide a deeper foundation to present the findings and conclusions of the study.

We have now described the model in more detail and described all the components of the model. New materials added can be found on p. 1-2  l. 40 - 70

 L.58-69. Two studies are introduced to provide detailed context for the study of PYD in youth footballers. This section would be enhanced if more specific information is provided about the context of each study. For example, adding the age-group, level of competition (e.g., club, competitive), location of the sample in each study.

We have provided respective details for these two studies p. 2 l. 80-89 and p. 2 l. 90-97

Results:

The findings are presented provide interesting insight about the developmental experiences of youth footballers. However, as Holt and colleagues model provided a meta-synthesis of existing knowledge about PYD in youth sport, I believe that the deductive analysis as presented does not provide the potential depth that it could. That is, only presenting the categories from the model provides confirmation of the model. The greatest insight from the findings centers around the ‘PYD outcomes’ and the ‘Negative Outcomes/Experiences’ as the themes appear to focus on the interaction between the individual athletes and the context. Could these be emphasized further to show how the findings inform specific components of the model?

The themes, as currently presented, highlight both positive and negative experiences and outcomes. Delineating these more explicitly would help to show what/how PYD occurs in the youth football context

We have added new materials in these two sections to describe in more detail the results. Further,  in the conclusion we have recognized that the deductive analysis may have restricted the depth of the description.

L. 197- How was the ‘Sport Club’ theme identified in the deductive framework? Does this represent the ‘Distal Ecological System’?

In the results section we explain how this theme was identified in the deductive framework (p. 4, l. 203-205)

In the discussion, we have added new materials to discuss how this theme fit in “Distal Ecological System” component of the Holt et al model.

L. 215 - The ‘Life Skills’ theme and sub-themes presented specific developmental outcomes, that in my view, would align more clearly under the ‘PYD outcomes’ theme? Holt et al., present the implicit process and the explicit process, with an emphasis on the learning process. Explain this process in addition to the specific PYD outcomes would be helpful to realign this section to the model may help to show this connection.

Thank you for this constructive comment. Indeed, Holt et al. emphasize the respective learning processes in the component Life Skills of the model. This is what we sought to describe in this section of the paper and that was the focus in the respective section in the interview. To this end, we have asked the participants to comment on the process of development or not of specific life-skills. We have inserted additional materials to describe such focusing on p. 8 l. 548-552. Further, we put respective youngsters’ reports in the context of life-skills learning processes categorization of Bean and Forneris (2017) (p. 8 l 554-560). We have also rephrased the 2nd RQ (p. 4, L. 202-203) to reflect the focus on the processes life-skills development is achieved.  Finally, to emphasize our focus on the process we have edited the name of the higher-order theme, from Life-skills to life-skills development.

Based a) on the above reasoning and respective edits and b) on the different focus of the interview regarding PYD outcomes, where we questioned participants about attributes possibly improved through football, we have opted to retain the different sections of Life-skills development and PYD Outcomes.

The authors state the ‘young athletes have a “naïve” view of life skills” (L.218-219). This is an interesting statement that could be expanded on – does this mean that life skills are not being learned to the degree explained in the literature?

We have added new materials in the discussion to explain this notion. (p. 12 – l. 1035-1048)

Discussion:

The authors describe “two types of coaches” (L. 343-). This highlights an interesting finding that adds to the literature of coaching for PYD. However, it would have been beneficial to present this more clearly in the results section to clearly delineate, with quotes and themes, how the two types of coaches behaviorally differed.

Thank you for this constructive comment. This prompted us to examine again our data and confirm that those coaches with a learning focus were also those who created a positive climate while those coaches with a winning focus were in most cases those that exhibited negative behaviors. Thus, we made this point in the respective results section (p 5, l 398-404) and we re-arranged the respective themes in Table 1 to reflect this distinction more clearly. We have also inserted additional respective quotes in this section, 

The authors state that the explicit process was not mentioned by participants (L. 368-). Does this infer that the process is not necessary and that implicit development of life skills occurs? What recommendations can be taken from such findings? Adding the specific practical recommendations for coaches would add depth to the paper.  

We have followed this comment and added new respective materials in the discussion (p. 12, l. 1038-1048)

Methods:

The choice of a phenomenological approach is based on an ontology of multiple perceptions of reality and an epistemology of subjectivity of human knowledge, yet is presented in a way that emphasizes “common meaning” (L. 411) and aligned with a deductive, theory-driven analysis. This appears to be an odd pairing of methodological approaches and a lack of methodological coherence. Can the authors explain and clarify the fit of these approaches?

Thank you for this comment. Our purpose was to provide a description of the common meaning of the lived experience of the young athletes. Although, a deductive theory driven analysis was chosen, we felt that regarding each component of the model (for example, interactions with coaches or interactions with parents) we could summarize students experiences and perceptions. However, we acknowledge that this could be an odd pairing of methodologies and therefore we opted eliminate the respective statement.

Adding detail about the timing/length of the interviews would be helpful.

We have added respective details

Round  2

Reviewer 2 Report

I commend the author/s on their thorough revisions and appreciate the detailed rationale and responses to the comments in the first review. While there are a couple of minor referencing and grammatical errors to be addressed (e.g., L. 78 - "3" should read "three"; L. 451 - "commended" should read "commented"), I am happy to recommend the paper for publication.